# The impact of COVID-19 lockdown on nursing higher education at Chengdu University

**Peiling Cai[1]ᵒ, Ya Shi[2]ᵒ, Jianing Cui[2], Junren Wang[2], Juan Ren[2], Brett D. Hambly[4], Shisan Bao**  **[1,3]\*, Zhongqing Xu[3]\***

1 School of Preclinical Medicine, Chengdu University, Chengdu, China, 2 Postgraduate Office, Chengdu University, Chengdu, China, 3 Department of General Practice, Tongren Hospital, Shanghai Jiaotong University School of Medicine, Shanghai, China, 4 Centre for Healthy Futures, Torrens University Australia, Sydney, New South Wales, Australia

ᵒ These authors contributed equally to this work.
\* zhongqing_xu@126.com (ZX); profbao@hotmail.com (SB)

## Abstract

### Background

To combat/control the COVID-19 pandemic, a complete lockdown was implemented in China for almost 6 months during 2020.

### Purpose

To determine the impact of a long-term lockdown on the academic performance of first-year nursing students *via* mandatory online learning, and to determine the benefits of online teaching.

### Methods

The recruitment and academic performance of 1ˢᵗ-year nursing students were assessed between 2019 [prior to COVID-19, n = 195, (146 women)] and 2020 [during COVID-19, n = 180 (142 women)]. The independent sample *t* test or Mann-Whitney test was applied for a comparison between these two groups.

### Results

There was no significant difference in student recruitment between 2019 and 2020. The overall performance of the first-year students improved in the Biochemistry, Immunopathology, Traditional Chinese Medicine Nursing and Combined Nursing courses *via* mandatory online teaching in 2020 compared with traditional teaching in 2019.

### Conclusion

Suspension of in-class learning but continuing education virtually online has occurred without negatively impacting academic performance, thus academic goals are more than achievable in a complete lockdown situation. This study offers firm evidence to forge a path for developments in teaching methods to better incorporate virtual learning and technology in order to adapt to fast-changing environments. However, the psychological/psychiatric

**Data Availability Statement:** All relevant data are within the paper and its Supporting Information files.

**Funding:** This research was supported by grants from First-class Curriculum Project of Chengdu

University in 2020 and 2021 (CDYLKC2021063 and CDYLKC2020016). This research was also supported by grants from Teaching Reform Projects of Chengdu University in 2022 (cdjgb2022060). This work was supported by grants from: Commission of Health, Changning District, Shanghai (YXMZK009, YZJH005, YZJH003), China Hospital Development Institute, Shanghai Jiao Tong University (CHDI-2020-A-26), Shanghai Association of Chinese Integrative Medicine (2020-12).

**Competing interests:** The authors have declared that no competing interests exist.

and physical impact of the COVID-19 lockdown and the lack of face-to-face interaction on these students remains to be explored.

## Introduction

The COVID-19 pandemic, that originally starting in December 2019, has severely impacted almost everyone around the world on physical, psychological and financial levels with unacceptably high rates of mortality (6,047,653) and morbidity (458,479,635) of COVID-19 to 16 March 2020 [1]. In order to control the transmission of COVID-19, several measures were implemented, including a complete lockdown of factories, schools and shopping malls, except for essential groceries, social distancing and mandatory facial mask wearing [2]. We have previously reported that there were COVID-19 transmissions from Wuhan to Gansu (Central to Northwest part of China) [3] and also reverse transmissions of COVID-19 from hotspots in Iran to Gansu Province in March 2020 [4]. Despite extreme precautions being taken to attempt to control the spread of COVID-19 in China, there have been outbreaks of the mutated COVID-19 virus, including the delta and more recent omicron strains [5], reported sporadically over the last six months. More recently, the omicron strain has been spreading quickly across numerous cities/provinces in China [6]. The biggest challenge in detecting the omicron strain is that it is more transmittable but less symptomatic in nature [6], i.e. many people become viral carriers without obvious clinical presentations, particularly among people who have had three vaccinations [7]. There are significant health, financial and social impacts of the COVID-19 pandemic that are yet to be determined, particularly in Shanghai, being the largest city in China with a population of 24 million [8].

To combat one of the worst pandemics to occur this century, the Chinese government implemented several restrictions, including a complete lockdown across the whole country, incorporating shutting down shopping malls except for essential groceries [3, 4], and restricting people's movements outdoors even within their communities. All schools were forced to suspend in-person learning for over 6 months, and virtual teaching became mandatory in almost every school or university in China in 2020 [9]. Following the re-outbreak of the current omicron strain of COVID-19, suspension of in-person learn has been re-introduced in many places in 2022 [10]. It is considered that the stress of the COVID-19 pandemic may have impacted the performance of higher education students, in particular due to the disruption to their learning as a result of the complete lockdowns, suspension of in-person learning and almost all laboratory-based practices. All Chinese schools and universities were mandatorily closed during the first and second waves of the COVID-19 outbreak [4] as well as the most recent omicron wave [6]. Various educational institutions responded to the COVID-19 restrictions differently [11], for example, transitioning from face-to-face teaching to online teaching where possible. For higher educational institutions, the key focus of the effects of the pandemic has been determining the impact on students following the switch to predominantly online teaching, particularly in relation to academic outcomes and performance. Mandatory online teaching due to lockdown has been studied amongst medical students has been studied previously, demonstrating promising outcomes for these medical students [12]. However, it remains to be explored what the impact of lockdown has been on nursing students.

It has been reported that mental health has been significantly harmed by the pandemic, e.g. increased stress levels, including fear of infection and isolation following long-term lockdown [13]. In the general population, there has been significantly increased depression among

young adults aged between 20–30 years undertaking a college or higher education, particularly among females [14]. Additionally, about 20% of medical staff and medical students have experienced increased anxiety and depression [15], perhaps due to excessive media consumption that has been linked to mood disorders during the lockdown in China.

However, the impact of lockdown on the training of nursing students remains to be explored. The importance of effective training of nursing students is well known [16]. These nursing students will become specialized registered nurses and play a critical role in the delivery of medical services either in hospitals and/or private practices. On the other hand, incompetent graduate nurses are likely to compromise health care delivery at various levels of outcomes for the management of patients.

Following the previous findings of a study of first-year medical students required to undertake online learning due to COVID-19, we would like to study the impact of online learning on first-year nursing students in a similar environment, as a direct comparison. The results and conclusions drawn from the current study will be useful in developing, organising and adapting future curriculums with the aid of technology for different areas of study, with and/or without a pandemic.

## Methods

Data focusing on the academic performance for first-year nursing students enrolled in 2018 (the pre-COVID-19 cohort) compared to those enrolled in 2019 (the COVID-19 cohort) were acquired from the administration data system of the School of Preclinical Medicine, Chengdu University, Chengdu, China. This research has been approved by the Ethics Committee, School of Preclinical Medicine, Chengdu University, in accordance with The Code of Ethics of the World Medical Association (Declaration of Helsinki) for experiments involving humans and the requirement for consent was waived by the Ethics Committee. None of the students included in the current study was a minor. The approval number is 2020–192 CDUMS, the date was 11/11/2020.

This study was an observational retrospective study. The study participants were from 1st year Nursing students in the Preclinical School of Chengdu University. All research data including students' gender, age and academic performance were based on electronic teaching management system of Chengdu University. In the current study, convenience sampling was adopted. There were 195 students in the pre-COVID-19 cohort, including 49 men and 146 women. There were 180 students in the COVID-19 cohort, consisting of 38 men and 142 women. None of the students included in the current study was a minor.

The nursing degree at Chengdu University is a 4-year program which consists of three parts, being Foundation Nursing Studies for the first 2 years, Clinical Studies for the following 1 year and a clinical internship in the Affiliate Hospital of Chengdu University in the final year. There are two semesters in each academic year. The first semester of the program (Autumn semester) runs between September and January each year and the second semester (Spring semester) runs from March to July. The pre-COVID-19 cohort undertook their first semester of study between September 2018 and January 2019 while the COVID-19 cohort undertook their first semester of study between September 2019 and January 2020, immediately prior to the onset of the pandemic and subsequent lockdown that occurred between February and July 2020 (Fig 1). Thus, the first semester of study for both cohorts was unaffected by the COVID-19 pandemic and lockdown. However, while the pre-COVID-19 cohort undertook their second semester during March to July 2019 prior to the onset of the pandemic, the COVID-19 cohort undertook their second semester during the first COVID-19 lockdown from March to July 2020.

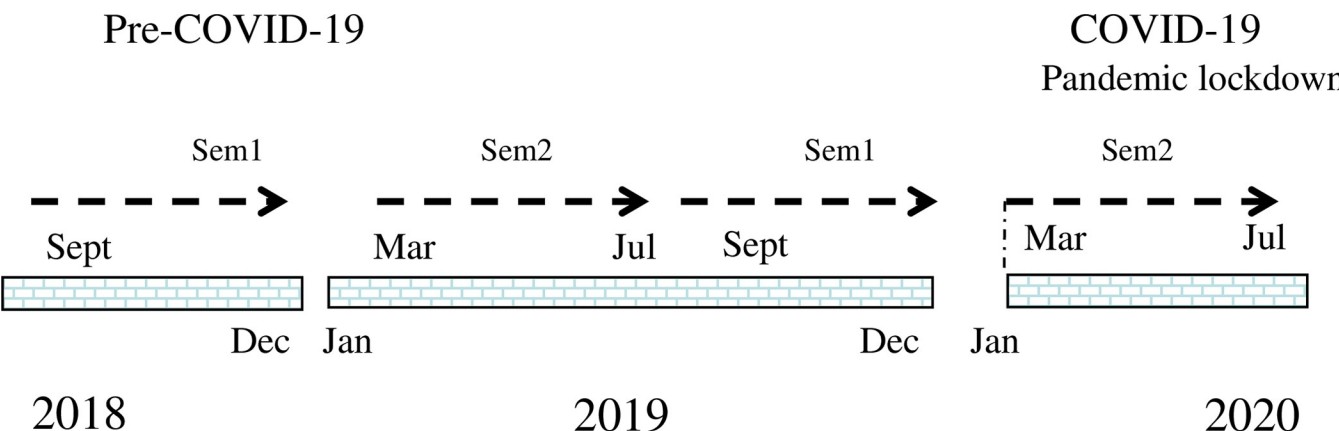

**Fig 1. Diagrammatic demonstration of the experimental design.** Chinese education is divided into two semesters each year. Semester 1 (Autumn semester) runs from September to the following February. Semester 2 (Spring semester) runs from March to July. During the pandemic lockdown between March and July 2020, the COVID-19 cohort carried out their second semester of studies online.

Ten pre-clinical subjects were undertaken during the second semester of the nursing program (March to July for 2019 and 2020): Chinese Classic Literature, English, Chinese History, Computer Science, Physiology, Biochemistry, Immunopathology, Physiology Practical, Traditional Chinese Medicine for Nursing (TCM Nursing) and Combined TCM and Western Nursing Course Training (Combined Training). Furthermore, the two cohorts carried out their final tests in the same way, that is face-to-face paper-based tests, when the students returned to the university after the outbreak. The performances of these two cohorts were evaluated as a whole and were further stratified by gender.

The contents of five of these ten subjects (Chinese Classic Literature, English, Computer, Chinese History and Combined Training) were presented entirely by lecture formats in both 2019 and 2020. However, in 2019, these lectures were presented once only in a timetabled face-to-face format, with only the PowerPoint content of the lecture being made available to the students online. In 2020, due to the pandemic, lectures in these subjects were mandatorily lived-streamed to all students at a fixed time. Additionally, in 2020, the presentations of all these lectures were recorded and made available online by the university authorities, allowing students to review the lecture content as often as required.

Importantly, the Biochemistry, Immunopathology and TCM Nursing courses consisted of both timetabled in person lectures and laboratory-based practical sessions. Historically, and in 2019, Biochemistry tuition was split about 79% between lectures (38 hours) and in-person practical sessions (10 hours); Immunopathology was split 75% between lectures (42 hours) and in-person practical parts (14 hours); and TCM Nursing was split 87.5% between lectures (28 hours) and in-person practical sessions (4 hours). However, in 2020, face-to-face lectures were replaced by online lectures, and in-person practical laboratory sessions were replaced by live-streams of a presenter demonstrating the experimental operations. The practical component of the above three courses were incorporated into the overall assessment of the subject.

By contrast, it is particularly emphasized that in the case of Physiology and Physiology Practical, Physiological Practical was taught and assessed separately to the Physiology component. The Physiology component consisted entirely of lecture-format teaching. However, educational innovation was introduced into Physiology in 2015 by the introduction of the Massive Open Online Courses (MOOCs) teaching methodology [17], which aimed to improve student engagement and learning by providing the required learning material to students prior to the in-person learning sessions, using both written and lecture-style pre-recorded media. The in-

person learning sessions that replaced the previous lecture format then focused on live prob-lem-solving and addressing questions from the students.

In the case of Physiology Practical during 2019, students attended scheduled practical clas-ses and mainly carried out experiments on animals, recording changes in physiological data, such as heart rate and respiration, resulting from external stimuli. PowerPoint presentations and other written materials for this subject were made available to students online during 2019. By comparison, during 2020, all teaching activities were conducted online with all mate-rial accessed online. Meanwhile, in-person practical sessions in the laboratory were replaced by online demonstrations of experimental operations. Students then completed the practical experimental operations based on online videos after returning to university in the following semester. Furthermore, the two cohorts carried out their final tests in the same way, that is using face to face onsite (paper based) tests, when the students returned to the university late in 2020 after the outbreak and lockdown.

## Statistical analysis

The Shapiro-Wilkerson test was utilized to evaluate whether the scores of each curriculum conformed to a normal distribution and to further compare the two sets of data. The quantita-tive information that adhered to a normal distribution was represented statistically using the mean and standard deviation ($\bar{x} \pm s$) and the independent sample $t$ test (homogeneous vari-ance) or $t'$ test (uneven variance) was used as a comparison between these two sets. For contin-uous variables that deviated from the normal distribution, that is, that showed a skewed distribution, these data were expressed using the median and interquartile range [M ($P$25, $P$75)] and the Mann-Whitney U test was applied to test the association between the two groups. GraphPad Prism, version 9.0, was employed for all statistical analyses. All statistical significance tests were set at two-sided, and a $P$ value of less than 0.05 was considered significant.

## Results

### Student recruitments in 2019 and 2020

There was no significant difference between the total student recruitments in 2019 (prior to the pandemic) and 2020 (pandemic period) (Fig 2A). In both 2019 and 2020, there were three times the number of women nursing students compared to men students (Fig 2B), which likely reflects historical gender roles and expectations within the nursing profession. The average age of the pre-COVID-19 cohort was (20.1 ± 1.3) years, including (19.9 ± 1.2) years for men and (20.2 ± 1.4) years for women. Similarly, the average age of the COVID-19 cohort was (20.1 ± 1.5) years, consisting of (20.6 ± 1.7) years for men and (20.0 ± 1.5) years for women (Table 1). There was no significant difference in age (Fig 2C, Table 1) or gender (Fig 2D, Table 1) between 2019 and 2020.

Surprisingly, the overall score achieved by students in Chinese Classic Literature was over 1.1 times higher in 2020 compared to 2019 (Fig 3A, Table 2). It should be highlighted that this course was taught online only in 2020, with free access to all learning materials. There was no significant difference in scores between the man and women students in 2019 or 2020 (Fig 3B, Table 3). The performance of both men and women students significantly improved in 2020 compared to their performance in 2019, respectively (Fig 3B, Table 3). Similar patterns were observed in English courses when comparing the performances in 2019 and 2020 (Fig 3C, Table 2) and the performances of men versus women (Fig 3D, Table 3). A similar pattern was observed in Chinese History barring one exception; women students in both 2019 and 2020 scored significantly higher than men (Fig 3E and 3F, Tables 2, 3). Finally, there was no

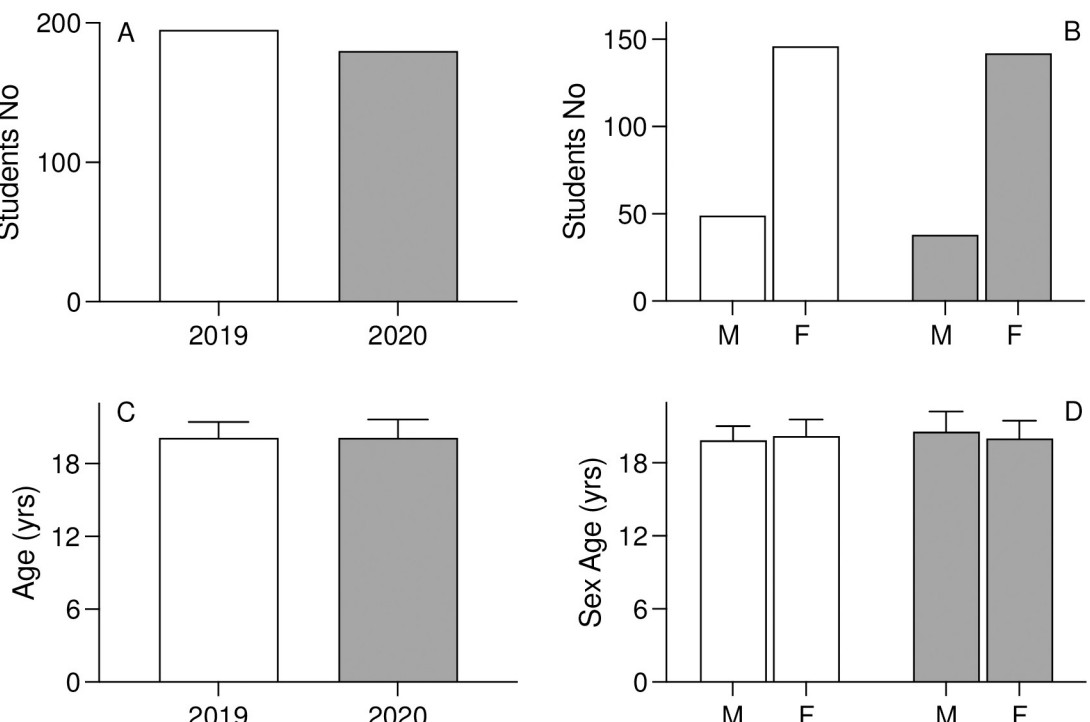

**Fig 2.** Equilibrium comparison of the students' demographic characteristics between the pre-COVID-19 cohort (white bars) and COVID-19 cohort (gray bars), including total number (A), numbers stratified based on gender (B), the average age (C) and the age stratified based on gender (D).

significant difference in the performance in Computer Science between 2019 and 2020 (Fig 3G, Table 2) or between men and women students (Fig 3H, Table 3).

The foundational course includes Physiology, Biochemistry, Pathology combined with Immunology (Immunopathology) and Physiology Practical. There was no difference of the performance in Physiology between 2019 and 2020 (Fig 4A, Table 4), nor between men and woman students in this subject (Fig 4B, Table 5). Surprisingly, the performance of Biochemistry declined significantly in 2020 compared to 2019 (Fig 4C, Table 4). However, the scores of woman students in Biochemistry were significantly higher than men students in 2019 (Fig 4D, Table 5). The overall performance of students in Immunopathology was significantly better in 2020 compared to the performance in 2019 (Fig 4E, Table 4). Women students outperformed men students in both 2019 and 2020. In addition, the women's performance was significantly better in 2020 than 2019 (Fig 4F, Table 5). Similar patterns to Immunopathology were observed in Physiology Practical (Fig 4G and 4H, Tables 4, 5).

**Table 1. Characteristics of the students between the pre-COVID-19 cohort and COVID-19 cohort.**

|  |  | Total Number | pre-COVID-19 cohort (n = 195) | COVID-19 cohort (n = 180) | $t$ / Chi-square test | |
|---|---|---|---|---|---|---|
|  |  |  |  |  | $t$ / $\chi^2$ value | $P$ value |
| Age (years) |  | 375 | 20.1 ± 1.3 | 20.1 ± 1.5 | 0.015 | 0.988 |
| Age (years) | men | 87 | 19.9 ± 1.2 | 20.6 ± 1.7 | 0.075 | 0.024 |
|  | women | 288 | 20.2 ± 1.4 | 20.0 ± 1.5 | 1.228 | 0.220 |
| Gender n (%) | men | 87 | 49 (25.1) | 38 (21.1) | 0.848 | 0.357 |
|  | women | 288 | 146 (74.9) | 142 (78.9) |  |  |

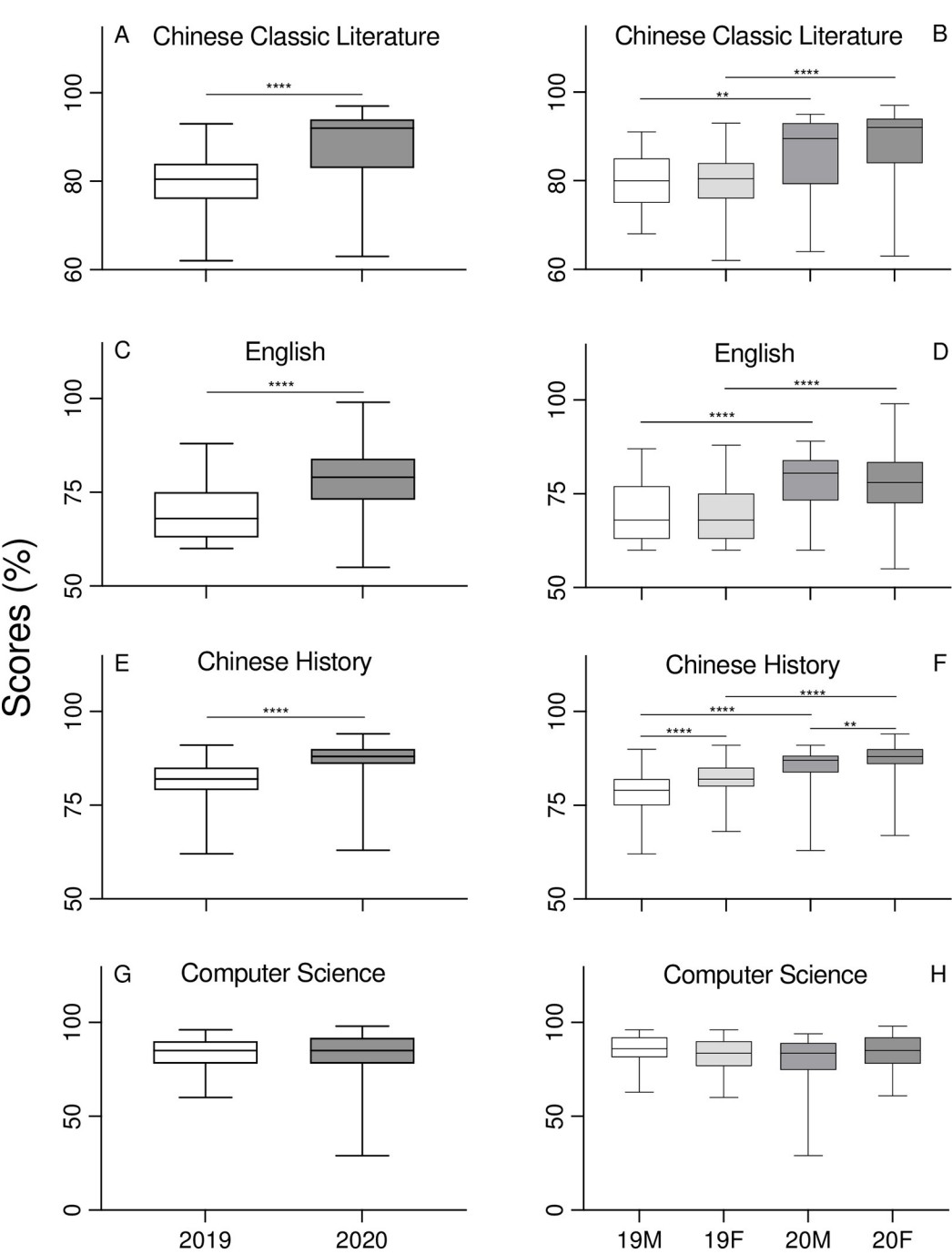

**Fig 3.** Student academic achievement by curriculum, Chinese Classic Literature (A), English (C), Chinese History (E) and Computer Science (G). The white and gray bars represent the pre-COVID-19 cohort and COVID-19 cohort, respectively. Student academic achievement of curriculum is further stratified by gender, Chinese Classic Literature (B), English (D), Chinese History (F) and Computer Science (H). The white, light grey, middle grey and dark grey bars represent men students in 2019, women students in 2019, men students in 2020 and women students in 2020, respectively. The Y-axis represents the academic mark in percentage points.

For the Traditional Chinese Medicine (TCM) course for nursing, there was significant improvement in the overall performance of all students in 2020 compared to 2019 (Fig 5A, Table 6). Interestingly, women only performed better than men in 2019 (Fig 5B, Table 7).

**Table 2. Academic achievements comparison of basic courses between the pre-COVID-19 cohort and COVID-19 cohort ($\bar{x} \pm s$).**

| Courses | pre-COVID-19 cohort (n = 195) | COVID-19 cohort (n = 180) | mean difference, 95% CI | t value | P value |
|---|---|---|---|---|---|
| Chinese Classic Literature | 80.21 ± 5.73 | 88.11 ± 8.23 | -7.90 (-9.36- -6.45) | -10.698 | <0.001 |
| English | 69.44 ± 7.55 | 77.39 ± 8.17 | -7.95 (-9.55- -6.35) | -9.780 | <0.001 |
| Chinese History | 81.25 ± 4.84 | 86.86 ± 5.16 | -5.60 (-6.62- -4.59) | -10.851 | <0.001 |
| Computer Science | 83.43 ± 8.12 | 83.19 ± 9.80 | 0.25 (-1.58–2.07) | 0.265 | 0.791 |

However, when compared by gender, both women and men in 2020 outperformed their 2019 counterparts, with a significant improvement in men from 2019 to 2020. The overall performance in the Combined Training curriculum was also significantly higher in 2020 than in 2019 (Fig 5C, Table 6). Further stratification is also illustrated by the similar pattern in the TCM and Combined Training course (Fig 5D, Table 7).

## Discussion

The current study has shown that the impact of the pandemic-driven change in teaching delivery to a purely online mode of teaching has not detrimentally impacted nursing student early learning, and, indeed, has substantially improved student learning in several subjects. Taken together, these data suggest that more extensive online learning methodologies should be incorporated into nursing training, although face-to-face learning still has a beneficial role in certain complex subjects, such as Biochemistry.

In the current study, we have demonstrated that the COVID-19 pandemic had no impact on the recruitment of nursing students by number, age and gender compared to recruitment prior to the pandemic. The number of women students was approximately three times higher than the number of men students in both 2019 and 2020. This may suggest there is a historic stigma associated with men in nursing which may be influencing the students' self-selection in the profession, as well as future requirements from hospitals in China. This is in line with other Asian countries, which may indicate that there are different perceptions and stigmas associated with nursing in Asia [18].

Student performance significantly improved in the subject of Chinese Classic Literature in 2020 with online teaching, compared to 2019 with the conventional lecture format. The improved performance was most likely due to a combination of factors. The COVID-19 lockdown-driven use of MOOCs [17] made the lectures and course materials more accessible to students and enabled them to review the materials as frequently as they wished in 2020. The complete lockdown may also have contributed to the improved score as it gave the students more time to focus on their studies with less distractions during the lockdown period in 2020,

**Table 3. Academic achievements comparison of basic courses between the pre-COVID-19 cohort and COVID-19 cohort based on gender stratification ($\bar{x} \pm s$).**

| Courses | Group | pre-COVID-19 cohort (n = 195) | COVID-19 cohort (n = 180) | mean difference, 95% CI | t value | P value |
|---|---|---|---|---|---|---|
| Chinese Classic Literature | Male | 80.40 ± 5.67 | 86.05 ± 9.12 | -5.66 (-9.04- -2.27) | -3.346 | 0.001 |
| | Female | 80.14 ± 5.77 | 88.66 ± 7.92 | -8.52 (-10.13- -6.91) | -10.405 | <0.001 |
| English | Male | 69.53 ± 7.70 | 78.13 ± 8.18 | -8.60 (-12.00- -5.20) | -5.028 | <0.001 |
| | Female | 69.41 ± 7.52 | 77.19 ± 8.18 | -7.77 (-9.60- -5.95) | -8.387 | <0.001 |
| Chinese History | Male | 78.31 ± 5.69 | 84.42 ± 7.03 | -6.11 (-8.83- -3.40) | -4.485 | <0.001 |
| | Female | 82.24 ± 4.09 | 87.51 ± 4.34 | -5.27 (-6.24- -4.29) | -10.606 | <0.001 |
| Computer Science | Male | 85.41 ± 7.66 | 81.21 ± 11.99 | 4.20 (-0.01–8.40) | 1.984 | 0.050 |
| | Female | 82.77 ± 8.19 | 83.72 ± 9.10 | -0.95 (-2.95–1.06) | -0.928 | 0.354 |

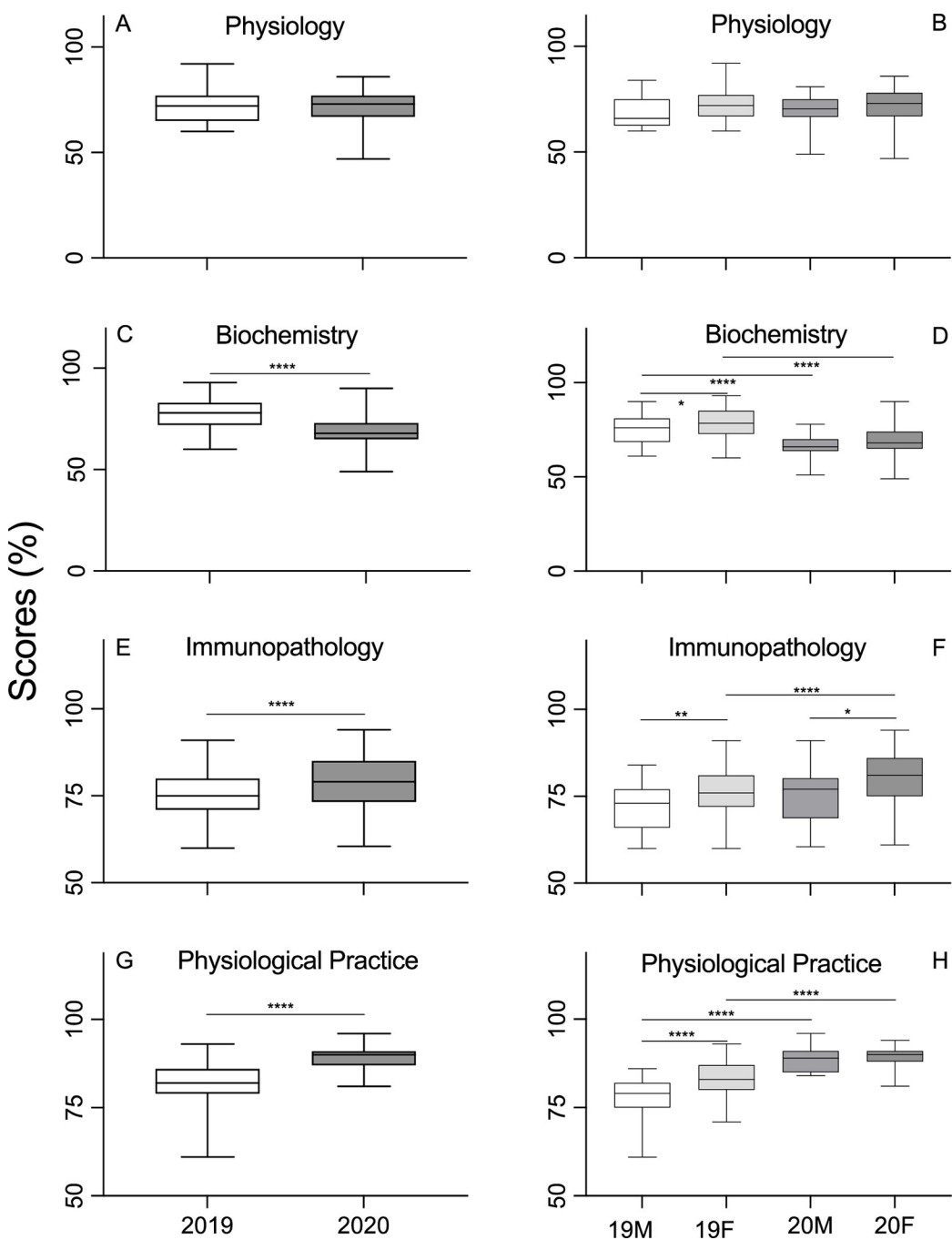

**Fig 4.** Student academic achievement by curriculum, Physiology (A), Biochemistry (C), Immunopathology (E) and Physiology Practice (G). The white and gray bars represent the pre-COVID-19 cohort and COVID-19 cohort, respectively. Student academic achievement of curriculum is further stratified by gender, Physiology (B), Biochemistry (D), Immunopathology (F) and Physiology Practice (H). The white, light grey, middle grey and dark grey bars represent the men students in 2019, women students in 2019, men students in 2020 and women students in 2020, respectively. The Y-axis represents the academic mark in percentage points.

compared to the same period in 2019. There was no significant difference in the scores in Chinese Classic Literature between men and women in either year, which may be due to the familiarity of the subject which was a continuation of study from high school.

**Table 4. Academic achievements comparison of foundational course between the pre-COVID-19 cohort and COVID-19 cohort ($\bar{x} \pm s$).**

| Courses | pre-COVID-19 cohort (n = 195) | COVID-19 cohort (n = 180) | mean difference, 95% CI | t value | P value |
|---------|-------------------------------|----------------------------|--------------------------|---------|---------|
| Physiology | 71.51 ± 7.42 | 71.75 ± 7.20 | -0.24 (-1.73–1.25) | -0.318 | 0.751 |
| Biochemistry | 77.52 ± 7.67 | 68.53 ± 6.67 | 8.99 (7.53–10.45) | 12.128 | <0.001 |
| Pathoimmunology | 74.94 ± 6.85 | 78.97 ± 8.06 | -4.02 (-5.55- -2.50) | -5.188 | <0.001 |
| Physiological Practice | 82.13 ± 4.96 | 88.92 ± 2.97 | -6.79(-7.61- -5.97) | -16.218 | <0.001 |

Similar patterns were observed in the English courses in 2019 and 2020, which could be explained by the same reasons as the patterns observed in the Chinese Classic Literature subject stated above. In China, English language learning begins in primary school in year 3, however, many students may have begun learning English even earlier as a result of "tiger parents" with extremely high expectations pushing their children to excel [19].

In Chinese History, women performed significantly better than men in both 2019 (traditional teaching) and 2020 (on-line teaching). This result may be attributed to the fact that women students are generally more mature than men between the ages of 18 and 25 years. This is supported by the report by Wu et al, which demonstrates that women have consistently outperformed men in tertiary medical courses [20]. Importantly, the general performance of this particular subject improved significantly in 2020 compared to in 2019. As explained above, these results illustrate the benefits of online teaching, which provides more flexibility for the students to review the course materials at a suitable time for them [17] as well as having more focused study time available during the complete lockdowns.

In the subject of Computer Science, there was no significant difference between the performance of men and women students in both 2019 and 2020, nor was there any material change in the overall performance between 2019 and 2020. As Computer Science is a logical curriculum which has historically involved considerable online teaching, the alternative approach of teaching using MOOCs for this subject is likely to be less affected by the adjustments to teaching methods as a result of the COVID-19 pandemic.

Surprisingly, there was no obvious difference in the scores in Physiology between men and women students in both 2019 and 2020, nor the general performance between 2019 and 2020. We speculate that the new students would have a general understanding of biology from their high school studies which may offer a bridge for the transition from high school biology to Physiology. Our speculation is supported by the finding that no significant difference was observed between men and women medical students in Physiology [21]. This explains the similar performance of both men and women in their Bachelor course and also the consistency of the overall scores between 2019 (traditional lectures) and 2020 (online teaching).

**Table 5. Academic achievements comparison of foundational course between the pre-COVID-19 cohort and COVID-19 cohort based on gender stratification ($\bar{x} \pm s$).**

| Courses | Group | pre-COVID-19 cohort (n = 195) | COVID-19 cohort (n = 180) | mean difference, 95% CI | t value | P value |
|---------|-------|-------------------------------|----------------------------|--------------------------|---------|---------|
| Physiology | Male | 69.20 ± 7.46 | 69.87 ± 7.41 | -0.66 (-3.86–2.53) | -0.413 | 0.680 |
| | Female | 72.28 ± 7.28 | 72.25 ± 7.09 | 0.29 (-1.64–1.70) | 0.035 | 0.972 |
| Biochemistry | Male | 75.22 ± 7.44 | 65.92 ± 6.00 | 9.30 (6.44–12.17) | 6.454 | <0.001 |
| | Female | 78.30 ± 7.62 | 69.23 ± 6.69 | 9.06 (7.40–10.72) | 10.733 | <0.001 |
| Pathoimmunology | Male | 71.96 ± 6.35 | 75.80 ± 7.55 | -3.84 (-6.81- -0.88) | -2.579 | 0.012 |
| | Female | 75.95 ± 6.74 | 79.82 ± 8.01 | -3.87 (-5.59- -2.16) | -4.441 | <0.001 |
| Physiological Practice | Male | 78.33 ± 5.04 | 88.58 ± 3.35 | -10.25 (-12.14- -8.37) | -10.811 | <0.001 |
| | Female | 83.40 ± 4.24 | 89.00 ± 2.86 | -5.60 (-6.44- -4.77) | -13.169 | <0.001 |

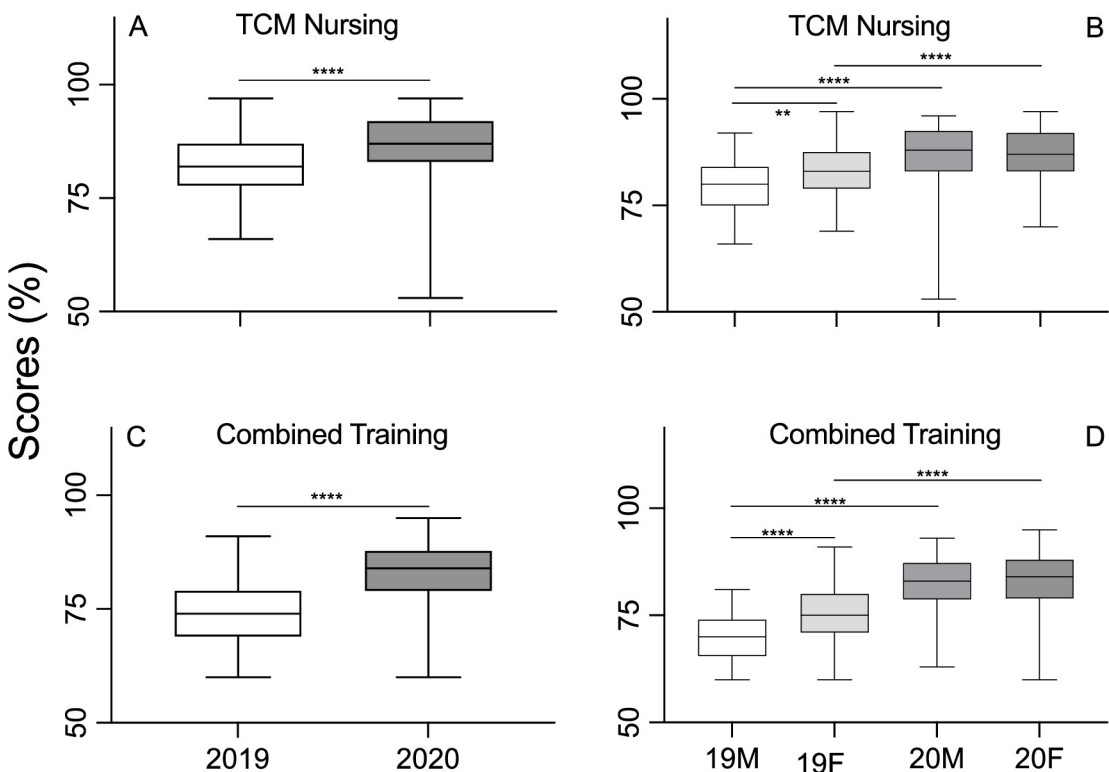

**Fig 5.** Student academic achievement by curriculum for TCM Nursing (A) and Multicultural Nursing (C). The white and gray bars represent the pre-COVID-19 cohort and COVID-19 cohort, respectively. Student academic achievement of curriculum is further stratified by gender, Chinese TCM Nursing (B) and Combined TCM and Western Nursing (D). The white, light grey, middle grey and dark grey bars represents the men students in 2019, women students in 2019, men students in 2020 and women students in 2020, respectively. The Y-axis represents the mark in percentage points.

By contrast, the scores in Biochemistry were significantly lower in 2020 during online teaching compared to that in 2019 with the traditional lecture format. Whilst to some extent it is an extension of chemistry, biochemistry is distinguishable from chemistry and the two subjects do not have the same overlap as biology and physiology. For this subject, students appeared to benefit more from in-person lectures than online recordings as it may facilitate a deeper understanding of the concepts. Such a finding suggests that there are some disadvantages to online teaching suggesting that tertiary education should not move to a complete virtual learning format. Existing traditional curriculums may be improved by developing a hybrid model that incorporates both in-person classes and online materials. For example, weekly online tutorials in addition to regular in-person lectures may assist students to provide a more flexible and supportive learning environment which may improve their performance. Our analysis is supported by the findings of Dumford and Miller which demonstrate that there are disadvantages to online teaching compared to traditional teaching methods [22].

**Table 6. Academic achievements comparison of nursing essential course between the pre-COVID-19 cohort and COVID-19 cohort ($\bar{x} \pm s$).**

| Courses | pre-COVID-19 cohort (n = 195) | COVID-19 cohort (n = 180) | mean difference, 95% CI | t value | P value |
|---|---|---|---|---|---|
| TCM Nursing | 82.08 ± 6.34 | 86.65 ± 7.05 | -4.56 (-5.93- -3.20) | -6.582 | <0.001 |
| Combined Nursing | 74.11 ± 7.16 | 82.77 ± 6.53 | -8.65 (-10.05- -7.26) | -12.201 | <0.001 |

**Table 7. Academic achievements comparison of nursing essential course between the pre-COVID-19 cohort and COVID-19 cohort based on gender stratification ($\bar{x} \pm s$).**

| Courses | Group | pre-COVID-19 cohort (n = 195) | COVID-19 cohort (n = 180) | mean difference, 95% CI | t value | P value |
|---|---|---|---|---|---|---|
| TCM Nursing | Male | 79.35 ± 5.90 | 86.08 ± 9.28 | -6.73 (-10.00- -3.47) | -4.103 | <0.001 |
| | Female | 83.01 ± 6.23 | 86.79 ± 6.38 | -3.78 (-5.25- -2.32) | -5.087 | <0.001 |
| Combined Nursing | Male | 69.80 ± 5.23 | 83.00 ± 5.95 | -13.20 (-15.59- -10.82) | -10.994 | <0.001 |
| | Female | 75.56 ± 7.15 | 82.70 ± 6.69 | -7.14 (-8.75- -5.54) | -8.750 | <0.001 |

In the course of Immunopathology, it has been demonstrated that online teaching produces superior performance compared to traditional lectures for the reasons explained above in relation to English and Chinese History. In the combined laboratory study, it is understood that the scores of women first-year students were better than their men counterparts in both 2019 and 2020 for the same reasons stated above. The 2020 cohort achieved higher scores than the 2019 cohort despite the 2020 student undertaking their studies online. This result was unexpected as the 2020 cohort was not able to gain first-hand laboratory practice for the practical course component. The cause of the improved performance in the combined laboratory practice from 2019 to 2020 remains unclear and is currently being investigated.

The overall scores for the TCM nursing course were significantly better in 2020 with the aid of online teaching, compared to the scores in 2019 with traditional lecturing. This finding further supports the advantages of flexible and efficient online teaching for nursing students. As expected, women students performed better than men students in 2019 but not in 2020. This may be due to the fact that both genders had similar study time under complete lockdown conditions, resulting in similar academic performances in 2020. Of course, the potential psychological impacts of the lockdown and the COVID-19 pandemic remain to be explored.

The results of the Combined TCM and Western Nursing Course Training reflected almost the same patterns as the standard TCM course, which is consistent with the overall hypothesis that online training is effective and efficient at producing desirable overall academic outcomes for at least new nursing students. Whilst Integrated Western Medicine and Traditional Medicine courses are mandatory in the Chinese medical education system [23], the proportion of Western Medicine and TCM courses varies depending on the nature of the degree. The purpose of this paper is not to discuss the advantages and disadvantages of such integration and the most beneficial ratio of the courses.

Notably, in several subjects, women students performed better than men students both pre-COVID-19 and/or during the lockdown. The students' performance might be influenced by a number of factors e.g. conscientiousness and self-motivation [24], which is consistent with the findings from others, showing that performance by women is better in tertiary health subjects [20]. The difference was thought to be a result of increased self-efficacy from women students. Thus, the perception from women students was that they possess a higher level of confidence, which motivated them to complete the required learning tasks for their course, and subsequently led to greater mental effort and persistence, compared to men students.

However, there are some limitations from the current study. Firstly, the current study focused on first-year nursing courses which are generally more foundational and theory-based than clinical or practical. We would have liked to extend our study to gathering data from second-year and third-year nursing courses, however, the course coordinators declined to provide these data. Secondly, the assessment was based on academic performance only and did not evaluate the physical and psychological impact of the pandemic and complete lockdown. A significant increase in the incidence of skeletomuscular pain among medical students has

been reported following remote online teaching during the COVID-19 pandemic in Jordan [25]. This suggests that there may be potential long-term physical impacts as a result of the limitation of proper physical exercise and excessive computer time. In addition, extended lockdowns and the stress of the COVID-19 pandemic has been proven by Simon *et al* to severely impact the psychological and psychiatric health of Australians, particularly among those with a history of mental health issues [26]. This is also supported by the results of our previous study which demonstrated that patients who visited general practice clinics immediately following the first complete lockdown in the first half of 2020 reported a disproportionate number of psychological issues [27]. Furthermore, it has been demonstrated that there has been significantly increased levels of depression and anxiety among Chinese medical students and other medical staff; while staff within the administrative economy have been highly stressed at the end of the lockdown in China [28]. These findings are supporting our current hypothesis that there is potential high risk contributing to impairment of psychological status of these students with serious and even potentially irreversible outcomes. These impacts of the COVID-19 pandemic will be explored in future studies.

## Conclusion

We conclude that the suspension of in-person learning but continuing education via online resources is more than achievable during a complete lockdown situation and can be implemented in a way to significantly benefit the academic performance of students. The results and conclusions from this study offer solid evidence for developing and improving future curriculums with a blend of traditional in-person and virtual teaching to benefit tertiary education, irrespective of a pandemic situation.

We further conclude that there is minimal negative impact on the first-year nursing students' academic performance as a result of the transition to virtual learning due to the COVID-19 pandemic. Furthermore, online teaching or MOOCs has been shown to be beneficial and has improved the academic performance of the first-year nursing students. Virtual teaching and utilising the technology available should be encouraged to continue and incorporated into the curriculums for higher education, even if the pandemic ends in the near future. Physical exercise is strongly encouraged to minimise the potential impact of COVID-19 restrictions, and psychological counselling should also be offered to these students to support their mental health.

## Supporting information

**S1 File.**
(PDF)

## Author Contributions

**Conceptualization:** Peiling Cai, Ya Shi, Shisan Bao, Zhongqing Xu.

**Data curation:** Peiling Cai, Jianing Cui, Junren Wang, Juan Ren.

**Formal analysis:** Ya Shi, Jianing Cui, Junren Wang, Juan Ren, Brett D. Hambly, Zhongqing Xu.

**Writing – original draft:** Peiling Cai.

**Writing – review & editing:** Peiling Cai, Brett D. Hambly, Zhongqing Xu.

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
