## [Decision Letter · Decision Letter 0]

12 Dec 2022

PONE-D-22-15585THE IMPACT OF COVID-19 LOCKDOWN IN NURSING HIGHER EDUCATION OF CHENDU UNIVERSITYPLOS ONE

Dear Dr.,

Thank you for submitting your manuscript to PLOS ONE. After careful consideration, we feel that it has merit but does not fully meet PLOS ONE’s publication criteria as it currently stands. Therefore, we invite you to submit a revised version of the manuscript that addresses the points raised during the review process.

We look forward to receiving your revised manuscript.

Kind regards,

Omar Mohammad Ali Khraisat, Associate Professor

Academic Editor

PLOS ONE

Journal Requirements:

“Nil”

Reviewers' comments:

Reviewer's Responses to Questions

**Comments to the Author**

1. Is the manuscript technically sound, and do the data support the conclusions?

Reviewer #1: Partly

Reviewer #2: Yes

Reviewer #3: Yes

2. Has the statistical analysis been performed appropriately and rigorously? 

Reviewer #1: I Don't Know

Reviewer #2: Yes

Reviewer #3: Yes

3. Have the authors made all data underlying the findings in their manuscript fully available?

Reviewer #1: No

Reviewer #2: Yes

Reviewer #3: Yes

4. Is the manuscript presented in an intelligible fashion and written in standard English?

Reviewer #1: Yes

Reviewer #2: No

Reviewer #3: Yes

5. Review Comments to the Author

Reviewer #1: The title

--- Capital letters are used in the title. The title should be corrected based on the journal's style.

Abstract:

-- Please explain more about the aim and method of the study in the abstract section.

-- What kind of statistical tests were used?

Introduction

---You have explained well the importance of complete lockdown in COVID-19, but have not said anything about the importance of proper training of the nursing students.

--- What is the difference between your study and previous studieshave been conducted in this field? Briefly explain in this section.

--- The limitations of prior research might also be mentioned by the authors as further support for their present investigation.

Methods

---Please write the type of study, sample size, data collection tool and sampling strategy

--- Mention the number of samples examined .

Results:

---Mention the age range of participants.

---Line 197: Was the method of taking the exam the same in both groups? This should be specified, for example, one group of face-to-face test and another group of online test can have an effect on the average score obtained in the two groups.

--- It should be specified that the performance score of the students in each course is calculated as an average score? If the average score is calculated, the average score of each course in two groups should be given in a table.

---Reporting the results in the form of a table to compare two groups based on demographic variables makes the work more beautiful and understandable.

--- It is better mention the possible score (range) for each subjects (Chinese Classic Literature, English, Computer, Chinese History and Combined Training,....) so easier to readers interpret the results.

Discussion

---The beginning of the discussion should be with the purpose of study

The researcher can look for more arguments and bring his own argument in the discussion. For example, maybe the reason for the difference in the number of male and female students is the effort and study of women to enter this field.

---In discussion explain the important results of similar studies and interpret the differences.

Conclusion

--- Explain more about the message you want to deliver in students support mental health.

Reviewer #2: I would suggest the authors to edit their work for grammar, consistency and in some pages, particularity at the conclusion to rewrite so that readers can get the meaning out of it. There is need to include the role of the overall study in a form of implication or recommendation.

Reviewer #3: The manuscript is written professionally, reading it was a pleasure. I have very minor comments to address:

- In the Abstract and Methodology sections the numbers of students (participants) in each cohort are not mentioned. The authors need to shed some light on this.

- Some of the paragraphs (under the Introduction section) are very short and can be merged. For example, the second and third paragraphs can be merged. This is simply because academic writing does not favor very short paragraphs.

- On the same page (pages are not numbered, but I assume page 9), line 100, it is mentioned that “Our University School of Medicine not did not decline”. Here “not” is repeated.

- Also, on the same page, line 102, it is mentioned that “(manuscript is currently being reviewed).”Here, check journal appropriate referencing style for referring to an under-review manuscripts.

- Conclusion section: Check spelling of the word ‘excise’ in the phrase “Physical excise” (line 349).

All the best to the authors.

6. PLOS authors have the option to publish the peer review history of their article (what does this mean?). If published, this will include your full peer review and any attached files.

Reviewer #1: No

Reviewer #2: **Yes: **Fekadu Mulugeta Asfaw

Reviewer #3: **Yes: **Suliman M. N. Alnasser

---

## [Author Response · Author response to Decision Letter 0]

16 Feb 2023

Reviewer #1: 

The title: Capital letters are used in the title. The title should be corrected based on the journal's style.

We have revised according to the Journal Style, it now reads: “The impact of COVID-19 lockdown in nursing higher education of Chengdu University”

Abstract:

-- Please explain more about the aim and method of the study in the abstract section.

We have modified the Abstract accordingly, it now reads: 

“Purpose: To determine the impact of a long-term lockdown on the academic performance of first year nursing students via online learning, and benefit of online teaching.

Methods: The recruitments and academic performance of 1st year nursing students were assessed between 2019 [prior to, n = 195, (49 men and 146 women)] and 2020 [during COVID-19, n = 180 (38 men and 142 women)]. The independent sample t test or Mann Whitney was applied for a comparison between these two sets.” 

-- What kind of statistical tests were used? 

To clarify this point, we have added the following sentences in the statistical analysis, it now reads: “The Shapiro-Wilkerson test was utilized to evaluate whether the scores of each curriculum conformed to a normal distribution and to further compare the two sets of data. The quantitative information that adhered to the normal distribution was represented statistically using the mean and standard deviation (x ®±s) and the independent sample t test (homogeneous variance) or t' test (uneven variance) was used as a comparison between these two sets. For continuous variables that deviated from the normal distribution, that is, that showed a skewed distribution, this data was expressed using the median and interquartile range [M (P25, P75)] and the Mann-Whitney U test was applied to test the association between the two groups.” (Materials and Methods: page 6, para 4, lines 197-198).

Introduction

---You have explained well the importance of complete lockdown in COVID-19, but have not said anything about the importance of proper training of the nursing students.

To address this point, we have added the following sentences in the Introduction, it now reads: “However, the impact of lockdown in the training of nursing students remains to be explored. The importance of training competent nursing students is well known, who will become specialized registered nurses and play a critical role in the medical practices either in hospitals and/or private practices. On the other hand, incompetent nurses would compromise health care delivery at various different levels of outcomes for the management of patients.” (Introduction, last para, line 103-107).

--- What is the difference between your study and previous studies have been conducted in this field? Briefly explain in this section.

To address this point, we have added the following sentences, it now reads: “It has been reported that the mental health has been significantly harmed by the pandemic, e.g. increased stress level, including fear of infection and isolation following long-term lockdown [12] among general populations, and anxiety among medical staff and medical students, perhaps due to excessive media consumption that was linked to mood disorders during the lockdown in China [13, 14].” (Introduction, page 3, para 4, line 104-108).

--- The limitations of prior research might also be mentioned by the authors as further support for their present investigation.

To address this point, we have added the following sentences, it now reads: “Online teaching has been introduced previously, demonstrating promising outcomes for medical and/or nursing students. The previous studies are focusing on general population and/or medical staff. However, it remains to be explored what the impact of lockdown is on nursing students. In addition, the online teaching in the previous study is a supplementary approach for face to face teaching, but not exclusive online teaching as presented in the current study.” (Introduction, page 3, para 2, lines 97-102).

Methods

---Please write the type of study, sample size, data collection tool and sampling strategy

--- Mention the number of samples examined.

To address this point, we have added the following sentences, it now reads: “This study was an observational retrospective study. The study participants were from 1st year Nursing students in Preclinical School of Chengdu University. All research data including students’ gender, age and academic performance were based on electronic teaching management system of Chengdu University. In the current study, convenience sampling was adopted. There were 195 students in pre-COVID-19 cohort, including 49 men and 146 women. And there were 180 students in COVID-19 cohort, consisting of 38 men and 142 women.” (Materials and Methods, page 5, para 1, line 129-135).

Results:

---Mention the age range of participants.

To address this point, we have added the following sentences, it now reads: “The age of pre COVID-19 cohort was (20.1 ± 1.3) years, including (19.9 ± 1.2) years for men and (20.2 ± 1.4) years for women. And the age of COVID-19 cohort was (20.1 ± 1.5) years, consisting of (20.6 ± 1.7) years for men and (20.0 ± 1.5) years for women (Table 1). (Results, page 7, para 1, lines 211-214).

Table 1 Characteristics of the students between the pre-COVID-19 cohort and COVID-19 cohort

 Total Number pre-COVID-19 cohort (n=195) COVID-19 

cohort (n=180) t / Chi-square test

 t / �2 value P value

Age (years) 375 20.1 ± 1.3 20.1 ± 1.5 0.015 0.988

Age (years) men 87 19.9 ± 1.2 20.6 ± 1.7 -2.307 0.024

 women 288 20.2 ± 1.4 20.0 ± 1.5 1.228 0.220

Sex n (%) men 87 49 (25.1) 38 (21.1) 0.848 0.357

 women 288 146 (74.9) 142 (78.9) 

---Line 197: Was the method of taking the exam the same in both groups? This should be specified, for example, one group of face-to-face test and another group of online test can have an effect on the average score obtained in the two groups.

To address this point, we have added the following sentences, it now reads: “Furthermore, the two cohorts carried out their final tests in the same way, that is offline tests, when the students returned to the university after the outbreak.” (Materials and Methods, page 6, para 3 lines 193-195).

--- It should be specified that the performance score of the students in each course is calculated as an average score? If the average score is calculated, the average score of each course in two groups should be given in a table.

We have modified accordingly, and presented the data in tables (Table 2-7).

---Reporting the results in the form of a table to compare two groups based on demographic variables makes the work more beautiful and understandable.

We have modified accordingly, and showed data in tables (Table 2-7).”

--- It is better mention the possible score (range) for each subjects (Chinese Classic Literature, English, Computer, Chinese History and Combined Training,....) so easier to readers interpret the results.

We have modified accordingly, and showed data in tables (Tables 2-7).

Table 2 Academic achievements comparison of basic courses between the pre-COVID-19 cohort and COVID-19 cohort (x ®±s)

Courses pre-COVID-19 cohort (n=195) COVID-19 cohort (n=180) mean difference,

 95% CI t value P value

Chinese Classic Literature 80.21 ± 5.73 88.11 ± 8.23 -7.90 (-9.36- -6.45) -10.698 <0.001

English 69.44 ± 7.55 77.39 ± 8.17 -7.95 (-9.55- -6.35) -9.780 <0.001

Chinese History 81.25 ± 4.84 86.86 ± 5.16 -5.60 (-6.62- -4.59) -10.851 <0.001

Computer Science 83.43 ± 8.12 83.19 ± 9.80 0.25 (-1.58- 2.07) 0.265 0.791

Table 3 Academic achievements comparison of basic courses between the pre-COVID-19 cohort and COVID-19 cohort based on sex stratification (x ®±s)

Courses Group pre-COVID-19 cohort (n=195) COVID-19 cohort (n=180) mean difference, 95% CI t value P value

Chinese Classic Literature men 80.40 ± 5.67 86.05 ± 9.12 -5.66 (-9.04- -2.27) -3.346 0.001

 women 80.14 ± 5.77 88.66 ± 7.92 -8.52 (-10.13- -6.91) -10.405 <0.001

English men 69.53 ± 7.70 78.13 ± 8.18 -8.60 (-12.00- -5.20) -5.028 <0.001

 women 69.41 ± 7.52 77.19 ± 8.18 -7.78 (-9.60- -5.95） -8.387 <0.001

Chinese History men 78.31 ± 5.69 84.42 ± 7.03 -6.11 (-8.83- -3.40） -4.485 <0.001

 women 82.24 ± 4.09 87.51 ± 4.34 -5.27 (-6.24- -4.29） -10.606 <0.001

Computer Science men 85.41 ± 7.66 81.21 ± 11.99 4.20 (-0.01- 8.40） 1.984 0.050

 women 82.77 ± 8.19 83.72 ± 9.10 -0.95 (-2.95- 1.06） -0.928 0.354

Table 4 Academic achievements comparison of foundational course between the pre-COVID-19 cohort and COVID-19 cohort (x ®±s)

Courses pre-COVID-19 cohort (n=195) COVID-19 cohort (n=180) mean difference, 95% CI t value P value

Physiology 71.51 ± 7.42 71.75 ± 7.20 -0.24 (-1.73- 1.25) -0.318 0.751

Biochemistry 77.52 ± 7.67 68.53 ± 6.67 8.99 (7.53- 10.45) 12.128 <0.001

Pathoimmunology 74.94 ± 6.85 78.97 ± 8.06 -4.02 (-5.55- -2.50) -5.188 <0.001

Physiological Practice 82.13 ± 4.96 88.92 ± 2.97 -6.79(-7.61- -5.97) -16.218 <0.001

Table 5 Academic achievements comparison of foundational course between the pre-COVID-19 cohort and COVID-19 cohort based on sex stratification (x ®±s)

Courses Group pre-COVID-19 cohort (n=195) COVID-19 cohort (n=180) mean difference, 95% CI t value P value

Physiology men 69.20 ± 7.46 69.87 ± 7.41 -0.66 (-3.86- 2.53） -0.413 0.680

 women 72.28 ± 7.28 72.25 ± 7.09 0.29 (-1.64- 1.70) 0.035 0.972

Biochemistry men 75.22 ± 7.44 65.92 ± 6.00 9.30 (6.44- 12.17) 6.454 <0.001

 women 78.30 ± 7.62 69.23 ± 6.69 9.06 (7.40- 10.72) 10.733 <0.001

Pathoimmunology men 71.96 ± 6.35 75.80 ± 7.55 -3.84 (-6.81- -0.88) -2.579 0.012

 women 75.95 ± 6.74 79.82 ± 8.01 -3.87 (-5.59- -2.16) -4.441 <0.001

Physiological Practice men 78.33 ± 5.04 88.58 ± 3.35 -10.25 (-12.14- -8.37) -10.811 <0.001

 women 83.40 ± 4.24 89.00 ± 2.86 -5.60 (-6.44- -4.77) -13.169 <0.001

Table 6 Academic achievements comparison of Nursing essential course between the pre-COVID-19 cohort and COVID-19 cohort (x ®±s)

Courses pre-COVID-19 cohort (n=195) COVID-19 cohort (n=180) mean difference, 

95% CI t value P value

TCM Nursing 82.08 ± 6.34 86.65 ± 7.05 -4.56 (-5.93- -3.20) -6.582 <0.001

Combined Nursing 74.11 ± 7.16 82.77 ± 6.53 -8.65 (-10.05- -7.26) -12.201 <0.001

 Table 7 Academic achievements comparison of Nursing essential course between the pre-COVID-19 cohort and COVID-19 cohort based on sex stratification (x ®±s)

Courses Group pre-COVID-19 cohort (n=195) COVID-19 cohort (n=180) mean difference, 

95% CI t value P value

TCM Nursing Male 79.35 ± 5.90 86.08 ± 9.28 -6.73 (-10.00- -3.47) -4.103 <0.001

 Female 83.01 ± 6.23 86.79 ± 6.38 -3.78 (-5.25- -2.32) -5.087 <0.001

Combined Nursing Male 69.80 ± 5.23 83.00 ± 5.95 -13.20 (-15.59- -10.82) -10.994 <0.001

 Female 75.56 ± 7.15 82.70 ± 6.69 -7.14 (-8.75- -5.54) -8.750 <0.001

Discussion

---The beginning of the discussion should be with the purpose of study

We have inserted an additional paragraph at the beginning of the Discussion summarising the results in terms of the stated aims, it now reads: “The current study has shown that the impact of the pandemic-driven change in teaching delivery to a purely online mode of teaching has not detrimentally impacted nursing student early learning, and, indeed, has substantially improved student learning in several subjects. Taken together, these data suggest that more extensive online learning methodologies should be incorporated into nursing training, although face-to-face learning still has a beneficial role in certain complex subjects, such as Biochemistry.” (Discussion, page 8, first para, 253-258) 

The researcher can look for more arguments and bring his own argument in the discussion. For example, maybe the reason for the difference in the number of male and female students is the effort and study of women to enter this field.

We apologise for this confusion. The discussion about difference between men and women students have been presented: “Surprisingly, there was no obvious difference in the scores in Physiology between male and female students in both 2019 and 2020, nor the general performance between 2019 and 2020. We speculate that the new students would have a general understanding of biology from their high school studies which may offer a bridge for the transition from high school biology to Physiology. Our speculation is supported by the finding that no significant difference was observed between male and female medical students in Physiology [19]. This explains the similar performance of both males and females in their Bachelor course and also the consistency of the overall scores between 2019 (traditional lectures) and 2020 (online teaching). (Discussion, page 8, last para, 300-307).”

The researcher can look for more arguments and bring his own argument in the discussion. For example, maybe the reason for the difference in the number of male and female students is the effort and study of women to enter this field.

---In discussion explain the important results of similar studies and interpret the differences.

To address this point we have added the following sentences in the Discussion, it now reads: “Notably, in several subjects, women students performed better than men students both pre-COVID 19 and/or during the lockdown. The students’ performance might be influenced by a number of factors e.g. conscientiousness and self-motivation [22], which is consistent with the findings from others, showing that performance by women is better in tertiary health subjects [18]. The difference was thought to be a result of increased self-efficacy from women students. Thus, the perception from women students was that they possess a higher level of confidence, which motivated them to complete the required learning tasks for their course, and subsequently led to greater mental effort and persistence, compared to men students.” (Discussion, page 9, last para 347-354) 

Conclusion

--- Explain more about the message you want to deliver in students support mental health.

Furthermore, it has been demonstrated that there has been significantly increased levels of depression and anxiety among the Chinese students and other medical staff; while economy staffs are highly stressed at the end of the lockdown in China [25]. These findings are supporting our current hypothesis that there is potential high risk contributing to impairment of psychological status of these students with serious and even potentially irreversible outcomes. These impacts of the COVID-19 pandemic will be explored in future studies. (Discussion, Page 10, first para, lines 370-375).

 

Reviewer #2

I would suggest the authors to edit their work for grammar, consistency and in some pages, particularity at the conclusion to rewrite so that readers can get the meaning out of it. There is need to include the role of the overall study in a form of implication or recommendation.

We have modified our manuscript accordingly. 

 

Reviewer #3

The manuscript is written professionally, reading it was a pleasure. I have very minor comments to address:

- In the Abstract and Methodology sections the numbers of students (participants) in each cohort are not mentioned. The authors need to shed some light on this.

We have modified accordingly, it now reads: “There were 195 students in pre-COVID-19 cohort, including 49 men and 146 women. And there were 180 students in COVID-19 cohort, consisting of 38 men and 142 women.”

- Some of the paragraphs (under the Introduction section) are very short and can be merged. For example, the second and third paragraphs can be merged. This is simply because academic writing does not favour very short paragraphs.

We have modified accordingly.

- On the same page (pages are not numbered, but I assume page 9), line 100, it is mentioned that “Our University School of Medicine not did not decline”. Here “not” is repeated

We have removed the extra "not". 

- Also, on the same page, line 102, it is mentioned that “(manuscript is currently being reviewed).” Here, check journal appropriate referencing style for referring to an under-review manuscripts.

We have replaced the published manuscripts with our manuscript under review, it now reads: “It has been reported that the mental health is significantly harmed, e.g. increased stress level, including fear of infection and isolation following long-term lockdown [12] among general populations, and anxiety among medical staffs and medical students, perhaps due to excessive media consumption was linked to mood disorders during the lockdown in China [13, 14]. However it remains to be explored the impact of lockdown in the training of nursing students. It is well known the importance of training competent nursing students, who would become specialized registered nurses and play critical role in the medical practices either in hospitals and/or private practices. On the other hand, incompetent nurses would compromise at the different levels of outcomes for the management of patients.” (Introduction, page 3, last para lines 104-112).

- Conclusion section: Check spelling of the word ‘excise’ in the phrase “Physical excise” (line 349).

We have modified the word, from ‘excise’ to ‘exercise’.

---

## [Decision Letter · Decision Letter 1]

11 Apr 2023

PONE-D-22-15585R1The impact of COVID-19 lockdown in nursing higher education of Chengdu UniversityPLOS ONE

Dear Dr. Bao,

Thank you for submitting your manuscript to PLOS ONE. After careful consideration, we feel that it has merit but does not fully meet PLOS ONE’s publication criteria as it currently stands. Therefore, we invite you to submit a revised version of the manuscript that addresses the points raised during the review process.

We look forward to receiving your revised manuscript.

Kind regards,

Omar M Khraisat, Associate Professor

Academic Editor

PLOS ONE

Journal Requirements:

Reviewers' comments:

Reviewer's Responses to Questions

**Comments to the Author**

1. If the authors have adequately addressed your comments raised in a previous round of review and you feel that this manuscript is now acceptable for publication, you may indicate that here to bypass the “Comments to the Author” section, enter your conflict of interest statement in the “Confidential to Editor” section, and submit your "Accept" recommendation.

Reviewer #3: All comments have been addressed

Reviewer #4: (No Response)

2. Is the manuscript technically sound, and do the data support the conclusions?

Reviewer #3: Yes

Reviewer #4: Yes

3. Has the statistical analysis been performed appropriately and rigorously? 

Reviewer #3: Yes

Reviewer #4: Yes

4. Have the authors made all data underlying the findings in their manuscript fully available?

Reviewer #3: Yes

Reviewer #4: Yes

5. Is the manuscript presented in an intelligible fashion and written in standard English?

Reviewer #3: Yes

Reviewer #4: No

6. Review Comments to the Author

Reviewer #3: Thank you for revising the manuscript. Although all my comments were addressed properly, I can spot very few language related issues. For example, in the Method section, the last sentence of the second paragraph reads: “No of the students included in the study…” (Lins 134-135). This needs rephrasing. Please double-check for language quality.

I wish you all the best.

Reviewer #4: • Introduction, last para, line 103-107: Consider providing a reference to support your claim that "training competent nursing students is well known." Furthermore, I recommend removing the word "competent" in front of the nursing students as this term may not accurately reflect their current level of expertise. Instead, you can consider using a more general term such as "training nursing students" or "developing nursing students' skills."

• Introduction, page3. Para 4 : “It has been reported that the mental health has been significantly harmed by the pandemic, e.g. increased stress level, including fear of infection and isolation following long-term lockdown [12] among general populations, and anxiety among medical staff and medical students, perhaps due to excessive media consumption that was linked to mood disorders during the lockdown in China. Please consider re-writing this part to make clearer. The sentence is too long and had poor sentence structure, which made it difficult to read and understand. The sentence could further be improved by specifying whose mental health is being referred to. Adding a subject would make the sentence more specific and help to clarify the intended meaning. For example, "It has been reported that the mental health of the general population has been significantly harmed by the pandemic.

• Similar to the above comment, please have another look at Introduction, page 3, para 2, lines 97-102). The authors write ‘The previous studies are focusing on general population and/or medical staff’. Please check the tense.

• Line 197: the authors write: To address this point, we have added the following sentences, it now reads: “Furthermore, the two cohorts carried out their final tests in the same way, that is offline tests, when the students returned to the university after the outbreak.” (Materials and Methods, page 6, para 3 lines 193-195). I am concerned about the use of ‘offline test’. While "offline" could suggest a face-to-face or onsite test, it is not explicit enough to remove ambiguity. Therefore, the authors could improve their response by specifying the nature of the test more clearly. Instead of using "offline," they could use "face-to-face" or "onsite" if that is indeed what they meant. Alternatively, they could describe the method of taking the exam more precisely, for example, by stating that the students took a paper-based test at the university.

• While reviewing your work, I noticed that there were some areas where clarity could be improved. Specifically, I believe that there may be instances where the language used is ambiguous or unclear, which could be causing confusion for readers. To address this issue, I would like to encourage you to take a more careful approach to your work, and to carefully read the entire manuscript to ensure clarity. It may be helpful to have a colleague or a professional editor review your work to identify any areas where clarity could be improved, and to provide suggestions for how to address these issues.

7. PLOS authors have the option to publish the peer review history of their article (what does this mean?). If published, this will include your full peer review and any attached files.

Reviewer #3: **Yes: **Suliman M. N. Alnasser

Reviewer #4: **Yes: **Isaac Amankwaa (PhD, RN)

---

## [Author Response · Author response to Decision Letter 1]

16 Apr 2023

Dear Dr Khraisat

We appreciate the constructive comments made by the reviewers, our responses are as follows: 

Review Comments to the Author

Reviewer #3, : Thank you for revising the manuscript. Although all my comments were addressed properly, I can spot very few language related issues. For example, in the Method section, the last sentence of the second paragraph reads: “No of the students included in the study…” (Lins 134-135). This needs rephrasing. Please double-check for language quality.

Dear Dr Alnasser, Thank you for the comments. We have had the manuscript edited by a native English speaking academic.

Reviewer #4, Isaac Amankwaa:

• Introduction, last para, line 103-107: Consider providing a reference to support your claim that "training competent nursing students is well known." 

The reference has been added. 

Furthermore, I recommend removing the word "competent" in front of the nursing students as this term may not accurately reflect their current level of expertise. Instead, you can consider using a more general term such as "training nursing students" or "developing nursing students' skills."

Agree, competent has been deleted, replaced with training nursing students. 

1. Introduction, page3. Para 4: “It has been reported that the mental health has been significantly harmed by the pandemic, e.g. increased stress level, including fear of infection and isolation following long-term lockdown [12] among general populations, and anxiety among medical staff and medical students, perhaps due to excessive media consumption that was linked to mood disorders during the lockdown in China. Please consider re-writing this part to make clearer. The sentence is too long and had poor sentence structure, which made it difficult to read and understand. The sentence could further be improved by specifying whose mental health is being referred to. Adding a subject would make the sentence more specific and help to clarify the intended meaning. For example, "It has been reported that the mental health of the general population has been significantly harmed by the pandemic.

We have clarified this text and added appropriate references, it now reads:“It has been reported that mental health has been significantly harmed by the pandemic, e.g. increased stress levels, including fear of infection and isolation following long-term lockdown [12]. In the general population, there has been significantly increased depression among young adults aged between 20-30 years undertaking a college or higher education degree, particularly among females [13]. Additionally, about 20% of medical staff and medical students have experienced increased anxiety and depression [14], perhaps due to excessive media consumption that has been linked to mood disorders during the lockdown in China.”

2. Similar to the above comment, please have another look at Introduction, page 3, para 2, lines 97-102). The authors write ‘The previous studies are focusing on general population and/or medical staff’. Please check the tense.

We have revised and clarified this statement, including correcting the tense, it now reads: “Mandatory online teaching due to lockdown has been studied amongst medical students has been previously, studied, demonstrating promising outcomes for these medical students (our ref). However, it remains to be explored what the impact of lockdown has been on nursing students.”

Line 197: the authors write: To address this point, we have added the following sentences, it now reads: “Furthermore, the two cohorts carried out their final tests in the same way, that is offline tests, when the students returned to the university after the outbreak.” (Materials and Methods, page 6, para 3 lines 193-195). I am concerned about the use of ‘offline test’. While "offline" could suggest a face-to-face or onsite test, it is not explicit enough to remove ambiguity. Therefore, the authors could improve their response by specifying the nature of the test more clearly. Instead of using "offline," they could use "face-to-face" or "onsite" if that is indeed what they meant. Alternatively, they could describe the method of taking the exam more precisely, for example, by stating that the students took a paper-based test at the university.

We have modified the text to clarify this point: “Furthermore, the two cohorts carried out their final tests in the same way, that is using face to face onsite (paper based) tests, when the students returned to the university late in 2020 after the outbreak and lockdown.”

3. While reviewing your work, I noticed that there were some areas where clarity could be improved. Specifically, I believe that there may be instances where the language used is ambiguous or unclear, which could be causing confusion for readers. To address this issue, I would like to encourage you to take a more careful approach to your work, and to carefully read the entire manuscript to ensure clarity. It may be helpful to have a colleague or a professional editor review your work to identify any areas where clarity could be improved, and to provide suggestions for how to address these issues.

We have had the manuscript edited by a native English speaking academic.

We have modified our manuscript according to the reviewers’ comments and hope the quality of our updated manuscript meets the publication standards of Plos One. Please let me know if you have any queries.

---

## [Decision Letter · Decision Letter 2]

15 May 2023

The impact of COVID-19 lockdown on nursing higher education at Chengdu University

PONE-D-22-15585R2

Dear Dr.,

We’re pleased to inform you that your manuscript has been judged scientifically suitable for publication and will be formally accepted for publication once it meets all outstanding technical requirements.

Kind regards,

Omar M Khraisat, Associate Professor

Academic Editor

PLOS ONE

Additional Editor Comments (optional):

Reviewers' comments:

Reviewer's Responses to Questions

**Comments to the Author**

1. If the authors have adequately addressed your comments raised in a previous round of review and you feel that this manuscript is now acceptable for publication, you may indicate that here to bypass the “Comments to the Author” section, enter your conflict of interest statement in the “Confidential to Editor” section, and submit your "Accept" recommendation.

Reviewer #4: All comments have been addressed

2. Is the manuscript technically sound, and do the data support the conclusions?

Reviewer #4: Yes

3. Has the statistical analysis been performed appropriately and rigorously? 

Reviewer #4: Yes

4. Have the authors made all data underlying the findings in their manuscript fully available?

Reviewer #4: Yes

5. Is the manuscript presented in an intelligible fashion and written in standard English?

Reviewer #4: Yes

6. Review Comments to the Author

Reviewer #4: I have had the opportunity to review the revisions submitted by the authors following my initial review.

I am pleased to note that the authors have addressed all of the points I raised in my initial review. Their responses were detailed, clear, and they have made appropriate revisions to the manuscript. The issues previously noted have been resolved adequately, which has significantly improved the quality of the paper. The revised manuscript now presents the research in a clear and coherent manner, with a well-structured argument and thorough analysis. The methodology is robust, and the conclusions drawn are well-supported by the evidence provided.

Given these improvements, I believe the paper adds substantial value to the current body of knowledge in this field. Therefore, I am recommending the manuscript for publication in its current form.

Thank you for the opportunity to review this work. I am looking forward to seeing it published in our esteemed journal.

7. PLOS authors have the option to publish the peer review history of their article (what does this mean?). If published, this will include your full peer review and any attached files.

Reviewer #4: **Yes: **Amankwaa Isaac

---

## [Editor Report · Acceptance letter]

9 Jun 2023

PONE-D-22-15585R2 

The impact of COVID-19 lockdown on nursing higher education at Chengdu University 

Dear Dr. Bao:

I'm pleased to inform you that your manuscript has been deemed suitable for publication in PLOS ONE. Congratulations! Your manuscript is now with our production department. 

Kind regards, 

on behalf of

Dr. Omar M Khraisat 

Academic Editor

PLOS ONE